# Enhanced Thermoelectric Performance of Polycrystalline Si_0.8_Ge_0.2_ Alloys through the Addition of Nanoscale Porosity

**DOI:** 10.3390/nano11102591

**Published:** 2021-10-01

**Authors:** S. Aria Hosseini, Giuseppe Romano, P. Alex Greaney

**Affiliations:** 1Department of Mechanical Engineering, University of California, Riverside, 900 University Avenue, Riverside, CA 92521, USA; shoss008@ucr.edu; 2Institute for Soldier Nanotechnologies, Massachusetts Institute of Technology, 77 Massachusetts Avenue, Cambridge, MA 02139, USA

**Keywords:** bulk thermoelectric, alloy thermoelectrics, porous thermoelectrics, nanoengineering

## Abstract

Engineering materials to include nanoscale porosity or other nanoscale structures has become a well-established strategy for enhancing the thermoelectric performance of dielectrics. However, the approach is only considered beneficial for materials where the intrinsic phonon mean-free path is much longer than that of the charge carriers. As such, the approach would not be expected to provide significant performance gains in polycrystalline semiconducting alloys, such as Si_x_Ge_1-x_, where mass disorder and grains provide strong phonon scattering. In this manuscript, we demonstrate that the addition of nanoscale porosity to even ultrafine-grained Si
${}_{0.8}$
Ge
${}_{0.2}$
 may be worthwhile. The semiclassical Boltzmann transport equation was used to model electrical and phonon transport in polycrystalline Si
${}_{0.8}$
Ge
${}_{0.2}$
 containing prismatic pores perpendicular to the transport current. The models are free of tuning parameters and were validated against experimental data. The models reveal that a combination of pores and grain boundaries suppresses phonon conductivity to a magnitude comparable with the electronic thermal conductivity. In this regime, *ZT* can be further enhanced by reducing carrier concentration to the electrical and electronic thermal conductivity and simultaneously increasing thermopower. Although increases in *ZT* are modest, the optimal carrier concentration is significantly lowered, meaning semiconductors need not be so strongly supersaturated with dopants.

## 1. Introduction

The performance of thermoelectric (TE) materials depends upon having both advantageous electrical transport properties, and low thermal conductivity, and is quantified by a dimensionless figure of merit, 
$ZT=\left(\sigma S^{2}\right)/\left(\kappa_{e}+\kappa_{l}\right)T$
. Here, 
$\kappa_{e}$
 is the electrical contribution to the thermal conductivity, 
$\kappa_{l}$
 is lattice thermal conductivity, 
σ
 is the electrical conductivity and *S* is the Seebeck coefficient (thermopower) [1,2,3,4,5]. In materials for which the mean-free path of charge carriers is much smaller than the mean-free path of heat carriers (phonons), a well-established approach to increasing 
ZT
 is to introduce nanoscale porosity. When tuned to the right length scale, scattering of phonons by pores can significantly reduce the heat carriers’ mean-free path with only a minor impact on electrical transport properties in the numerator of 
ZT
 [6,7,8,9,10,11,12,13,14,15,16]. This approach is particularly appealing because it can yield dramatic increases in energy conversion efficiency in materials such as silicon that have not traditionally been considered to be good thermoelectrics [6,17,18]. It thus opens the door for the creation of energy harvesting devices that are fabricated from inexpensive, abundant, and environmentally benign materials, making them intrinsically scalable. However, the approach is considered to offer little further benefit to most established high-performance thermoelectrics, such as Si_x_Ge_1-x_ alloys [19]. To achieve high 
ZT
, these materials already possess one or more mechanisms for strong scattering phonons in their bulk form [20,21,22], and so, the conventional wisdom is that there is a diminishing return on the effort and cost required to add more phonon scattering. In this manuscript, we present models of the phonon and electron transport in nanoporous Si_x_Ge_1-x_ alloys and use these to compute the full thermoelectric figure of merit as a function of the material’s morphology and carrier concentration. These models show that there can be benefits to adding porosity to even good thermoelectrics, such as Si_x_Ge_1-x_ and that these benefits result not just in improved 
ZT
, but also the potential for reduced cost and better tolerance to overheating and microstructural evolution.

The Si_x_Ge_1-x_ alloy system is a well-established material for high-efficiency thermoelectrics, which is used in many niche applications, such as the thermoelectric generators that power deep-space probes, where efficiency and reliability take precedence over cost [21]. The alloy forms a fully-miscible solid solution at all values of x. The mass disorder of the randomly-distributed heavy Ge atoms strongly scatters short-wavelength phonons. Furthermore, these alloys can be fabricated from hot-pressed powder compacts to create materials with ultrafine grain size. The grain boundaries provide strong scattering of long-wavelength phonons, and together, the combination of grain structure and mass disorder leading to a strong suppression of the lattice thermal conductivity and a large 
ZT
. The lowest thermal conductivities occur at compositions with x∼0.5 [23]. The electronic properties in the numerator of 
ZT
 are tuned independently of the phonon scattering by controlling the doping concentration, and to obtain the optimal power factor the doping typically must be supersaturated, which means that the thermoelectric performance of these alloys can be degraded if the material is accidentally heated to a temperature at which dopant becomes mobile and precipitates out of solution. Although Si and Ge are both non-toxic, unlike the components of other widely used high efficiency thermoelectric materials (such as PbTe [24,25] and SnSe [26,27]), a second drawback of Si_x_Ge_1-x_ thermoelectrics is their cost. The price per mol of germanium is roughly two orders of magnitude larger than silicon, and so to reduce the expense (and expand the economic viability) of Si_x_Ge_1-x_ thermoelectrics we would like to improve the efficiency of compositions containing a relatively low Ge fraction. For this reason, in this manuscript, we focus exclusively on the Si
${}_{0.8}$
Ge
${}_{0.2}$
 alloy composition.

In the sections that follow, we first describe calculations that solve the Boltzmann transport equations for phonons in polycrystalline Si
${}_{0.8}$
Ge
${}_{0.2}$
 containing nanoscale extended pores with different cross-sectional shapes. The section following that presents a semiclassical model of electrical transport in *n*-type Si_x_Ge_1-x_, along with models of electron scattering by pores and grain boundaries, which were used to compute the electrical conductivity, Seebeck coefficient, and Lorenz number. The final section examines the combination of these in the 
ZT
 and discusses the options available for tuning morphology and dopant to optimize it.

## 2. Thermal Transport in Nanoporous Si_0.8_Ge_0.2_

The effective lattice thermal conductivity, 
$\kappa_{l}$
, of polycrystalline Si_0.8_Ge_0.2_ containing nanoscale pores was computed by solving the frequency-dependent Boltzmann transport equation [28] to find the steady-state distribution of phonons moving between an array of pores under an imposed temperature gradient. The effective thermal conductivity of the material containing a given pore morphology is defined as the ratio of the heat flux carried by the phonon distribution divided by the imposed temperature gradient. These simulations were performed using the OpenBTE Boltzmann transport solver [29] making use of materials properties for Si_x_Ge_1-x_ computed from first principles. The development and validation of OpenBTE with experimental data has been established in the literature through a series of publications [30,31,32,33]. The model incorporated the effects from four different phonon scattering processes: three-phonon scattering, elastic mass impurity scattering, scattering from grain boundaries, and scattering from pores. While the latter of these was modeled as physical obstacles in the simulation domain, the first three scattering processes were modeled implicitly using the single relaxation time approximation with the combined scattering rate from the three processes obtained using Matthiessen’s rule.

The second and third-order interatomic force constants for bulk Si_0.8_Ge_0.2_, computed with density functional theory using the virtual crystal approximation, were obtained from the AlmaBTE materials database [34]. The phonon dispersion was computed from the second-order force constants on a 
40×40×40
 point Brillouin zone mesh using AlmaBTE. The scattering matrices for three phonon interactions were computed from the third-order force constants also using AlmaBTE [35], which computes the full three-phonon scattering matrix and uses it to solve the linearized Boltzmann transport equation for phonons [34]. This method does not account for correlations, local relaxations, or changes in electronic structure due to alloying and interatomic force constant disorder, yet gives reasonable prediction for bulk Si_0.8_Ge_0.2_ thermal properties [36]. The rate of elastic phonon scattering by disordered germanium atoms was modeled by treating the Ge as random mass perturbations with the scattering rate given by Tamura’s formula for isotopic scattering [37]. The phonon–phonon scattering rate is temperature-dependent, while phonon–alloy scattering rate is temperature-independent, and the phonon lifetime from their combined effect, which we refer to as 
$\tau_{bulk}$
, is plotted in Figure 1a.

The rate of scattering of phonons by grain boundaries was approximated by assuming that the average interval for a free-flying phonon to collide with a boundary is 
$\tau_{grain}=l_{g}/\nu_{g}$
, where 
$\nu_{g}$
 is the phonon’s group velocity and 
$l_{g}$
 is effective grain size. This model acknowledges that phonons with different wavevectors and polarizations have different speeds but assumes that all phonons behave the same when they encounter a grain boundary scattering, scattering diffusely. The model is well known to slightly overestimate the thermal resistance from grain boundaries [38,39,40]. Figure 1a shows the distribution of 
$\tau_{bulk}$
 in red, and 
$\tau_{grain}$
 in blue, computed at 500 K for polycrystalline Si_0.8_Ge_0.2_ with an effective grain size of 50 nm. The total lifetime obtained using Matthiessen’s rule is plotted in purple. It can be seen that grain boundaries only dominate the scattering of acoustic phonons with frequencies lower than ∼2 THz, while lattice (Umklapp and Normal) and alloy scattering dominate for phonons with frequencies higher than that. The phonon group velocities used in the calculation of 
$\tau_{grain}$
 are shown in Figure 1b.

The effect of grain size on the thermal conductivity of polycrystalline Si_0.8_Ge_0.2_ is plotted in Figure 2a, normalized by the thermal conductivity of the single crystal. For an effective grain size of 
$l_{g}=200\;\mathrm{nm}$
, the largest grain size considered, the thermal conductivity is ∼40% of the single crystal alloy at high temperature (>
1000K
) and close to ∼25% at room temperature. Reducing the effective grain size to 
$l_{g}=20\;\mathrm{nm}$
 further decreases the thermal conductivity to below ∼20% of the single crystal conductivity at high temperatures and less than ∼10% at room temperature.

The effect of phonon scattering from a square array of nanoscale pores was modeled in the transport simulations by explicitly resolving pore geometry in the simulation domain, with the pore/semiconductor interface modeled as a diffusely-scattering adiabatic boundary. This means that the total energy flux incident on an interface is re-emitted back into the simulation domain in all directions distributed over all ordinates and phonon frequencies in proportion to their equilibrium occupancy. The simulation domain was periodic in all directions, and the pores were prisms that extend through the periodic boundaries in one direction. Three pore geometries were considered: cylinders, square prisms, and triangular prisms. The spacing between pores was adjusted to study the effect of pore density on thermal conductivity, and the pore size was adjusted concomitantly to maintain a constant pore fraction of 
ϕ=0.25
—a pore fraction similar to that of the nanoporous Si films reported in experimental works [23]. As this pore fraction is relatively large, the smallest pore spacing considered was limited to 10 nm to ensure that the spacing between pores remained large enough that the confinement effects were not significant, and that the electron and phonon dispersion of the material in the ligature between pores could still be reasonably approximated by those of the bulk crystal.

Figure 2b shows the thermal conductivity of a single crystal Si_0.8_Ge_0.2_ film containing an array of nanopores with different shapes and spacing. The triangular pores yield the lowest thermal conductivity of the different geometries considered here. This is primarily due to the phonon view factor [41,42]—a detailed discussion on the effect of the pores’ shape is given in the Appendix A. The pore–pore distance is the governing factor in thermal conductivity reduction. For a given porosity, shortening the pore–pore spacing (and therefore increasing the number density of pores) lowers the thermal conductivity and makes it insensitive to temperature [31]. The increase in the thermal conductivity ratio of the 
0.1 μ
m spaced pores in Figure 2b is entirely due to the decrease of the thermal conductivity of single-crystal Si_0.8_Ge_0.2_ with increasing temperature. This is mainly due to the suppression of long mean-free path phonons near the Brillouin zone center—the main contributors to thermal conductivity in Si_0.8_Ge_0.2_

While the plots in Figure 2 show the separate effect on lattice thermal conductivity from grain boundaries and nanopores independently, the synergy of cylindrical pores and polycrystallinity is shown in Figure 3 for a variety of different grain and pore sizes. Although nanopores and grain boundaries both present obstacles for phonon scattering that reduce thermal conductivity relative to the monolithic single crystal, the introduction of voids also creates regions in the material where the thermal conductivity is locally zero—a composite of non-conducting fibers within a conductive matrix. In the diffusive limit, the thermal conductivity of this composite is described by effective medium theory and, for cylindrical pores, depends on the pore fraction as 
$\kappa_{\mathrm{composite}}=\frac{1-\phi }{1+\phi }\kappa_{\mathrm{matrix}}$
 [43]. From Figure 3 it can be seen that adding porosity to polycrystalline Si_0.8_Ge_0.2_
*always* further reduces the thermal conductivity. If the pore spacing is significantly larger than the effective grain size the reduction in thermal conductivity is simply that of effective medium limit, however, we start to see an extra reduction in thermal conductivity even for pore spacings that are several times larger than the effective grain size.

Returning our attention to the diffuse limit, we note that the effective medium limit is also seen in the calculations for porous single-crystal Si_0.8_Ge_0.2_ plotted in green in Figure 2b. In this plot, we see that, while the shape of the pores makes minimal difference to the thermal conductivity reduction in the ballistic limit, when the pore spacing is small, the pore shape does impact the effective medium theory limit. While cylindrical pores yield a 0.6 reduction in thermal conductivity, consistent with the equation above, the pores with triangular cross-section reduce thermal conductivity by ∼0.48 consistent with the 
$\kappa_{\mathrm{composite}}/\kappa_{\mathrm{matrix}}=1-4.37\phi^{3}+3.47\phi^{2}-2.67\phi$
 formula [44].

Figure 4 shows the contribution to thermal conductivity from each phonon mode across the frequency spectrum in single-crystal Si_0.8_Ge_0.2_ (Figure 4a), and how this changes with a 200 nm grain structure (Figure 4b), addition of 500 nm spaced cylindrical pores (Figure 4c), and the combination of both (Figure 4d). This shows that the longest mean-free paths in the pristine material are suppressed in structures with defects. Moreover, even though the pore spacing is more than double the distance between grain boundaries there is a significant further reduction in the mean-free path of the low-frequency modes when both pores and grain boundaries are present. One can also observe some mean-free path suppression for high-frequency modes with the addition of nanopores.

## 3. Charge Carriers Transport in Nanoporous Si_0.8_Ge_0.2_

In order to obtain good thermoelectric properties, in most cases, Si_x_Ge_1-x_ thermoelectrics must be doped to high carrier concentrations. This can require the material to be doped beyond its solubility limit, which makes that device’s properties easily degraded irreversibly if the material is overheated to a point where the dopant becomes mobile and can precipitate out of solution. For phosphorus-doped Si_x_Ge_1-x_, experiments have shown that the carrier concentration varies with temperature as the solubility of the P dopant changes [45]. The variation is more noticeable at temperatures above 1000 K. In the work that follows, we restrict our attention to heavily *n*-type Si_x_Ge_1-x_, such as is obtained by doping with phosphorus, and we study the interplay between electrical and heat transport properties as the nanostructure and carrier concentration are varied.

The electrical properties in many semiconductors are described well by the semiclassical Boltzmann transport equation using the single relaxation approximation, integrating the contribution to transport from the charge carriers over a single electronic band [46]. This method has been used successfully to predict the transport coefficients of Si_x_Ge_1-x_ [19,47]. In this model, the electrical conductivity, 
σ
, at temperature *T* is written as [46]

(1)
$$\sigma =-\frac{1}{3}e^{2}\int \chi \left(E,T\right)\tau \left(E,T\right)dE,$$

with 
τ(E,T)
 the momentum relaxation time of electrons with energy *E*. The kernel 
χ(E)
 includes all the intrinsic non-scattering terms and is given by

(2)
$$\chi \left(E,T\right)=\nu^{2}\left(E\right)D\left(E\right)\frac{df(E,E_{f},T)}{dE},$$

were 
$E_{f}$
 is the Fermi energy level, 
ν(E)
 is the charge carrier group velocity, 
$f(E_{f},E,T)$
 is the Fermi–Dirac distribution, and 
D(E)
 is density of electronic states.

The Seebeck coefficient, *S*, and charge carriers’ contribution to thermal conductivity, 
$\kappa_{e}$
, depend on higher moments of 
χ
 with

(3)
$$S=-\frac{1}{eT}\frac{\int \gamma \tau dE}{\int \chi \tau dE},$$

and

(4)
$$\kappa_{e}=-\frac{1}{3T}\left(\int \zeta \tau dE-\frac{{\left(\int \gamma \tau dE\right)}^{2}}{\int \chi \tau dE}\right).$$


Here the terms 
γ
 and 
ζ
 are energy weighted 
χ
 given by 
$\gamma =\chi \left(E-E_{f}\right)$
 and 
$\zeta =\chi {\left(E-E_{f}\right)}^{2}$
, respectively, and the explicit functional dependence of the terms has been dropped from the notation for compactness and clarity.

To evaluate the function 
χ
 in Equations (Equation 1)–(Equation 4) requires knowing the density of states, carrier group velocity, and Fermi energy. We modeled density of states of the Si_x_Ge_1-x_ conduction band, 
D(E)
, using the standard expression for a non-parabolic electron band

(5)
$$D\left(E\right)=\frac{m_{e}^{\frac{3}{2}}}{\pi^{2}\hbar }\left(1+2\alpha E\right)\sqrt{2E(1+\alpha E)},$$

where 
$m_{e}$
 is the electrons’ density of state effective mass (which is separate from the transport effective mass used later). For Si_x_Ge_1-x_ alloys the density of states is found to be well represented across a wide range of compositions using 
$m_{e}=\left[1.08(1-x)+1.41x-0.183x(1-x)\right]m_{o}$
, where 
$m_{o}$
 is free electron rest mass equal to 
$9.11\times 10^{-31}$
 kg [48], and with the anharmonicity term, 
α=0.5
 eV
${}^{-1}$
. This later term describes the deviation of the conduction band from a parabolic shape due to the admixture of an s-like conduction band states and p-like valence band states [49].

At compositions with less than 85% Ge, the band structure of Si_x_Ge_1-x_ matches that of Si [48], and so the electron group velocity was obtained from the slope of the conduction band along the conduction band valley in Si obtained from density functional theory (DFT). That is, 
$\nu =\frac{1}{\hbar }\left|\nabla_{k}E\right|$
 along the 
$\langle 100\rangle$
 directions on the 
Γ
 to *X* Brillouin zone path. The Si band structure was computed with the Vienna Ab initio Simulation Package (VASP) [50,51,52,53], and using the generalized gradient approximation (GGA) with the Perdew-Burke-Ernzerhof exchange-correlation functional (PBE) [54]. Projector-augmented wave (PAW) pseudopotentials were used to represent ion cores and their core electrons [55,56], and the Kohn–Sham wave functions were constructed using a planewave basis set with a 700 eV energy cutoff. A Monkhorst-Pack 
12×12×12
 k-point grid was used to sample the Brillouin zone [57]. The primitive cell and atomic basis were relaxed to minimize forces on the atoms to better than 10^−6^ eV/Å. The electronic band structure used to compute 
ν(E)
 was interpolated from a 
45×45×45
 k-point grid. Finally, the band structure, and, therefore, group velocity, were treated as temperature independent.

The final term that appears in 
χ
 is the Fermi energy. This term is not an intrinsic property and is strongly dependent on the carrier concentration and temperature. For a given carrier concentration, 
$n_{c}$
, the Fermi energy, 
$E_{f}$
, was computed self-consistently with the density of states in Equation (Equation 5) by numerically solving the integral equation

(6)
$$n_{c}=\int_{0}^{\infty }D\left(E\right)f\left(E,E_{f},T\right)dE,$$

using the conduction band edge to set the reference frame.

To complete the transport model, we need to compute the meantime between electron scattering events. In bulk Si_x_Ge_1-x_ the dominant electron scattering processes are scattering by acoustic phonons (
$\tau_{p}$
), ionized impurities (
$\tau_{i}$
) and alloy disorder (
$\tau_{a}$
). Ravich has modeled the rate of electron–phonon scattering as [58]

(7)
$$\tau_{p}{\left(E\right)}^{-1}=\frac{\pi D_{A}^{2}k_{B}TD\left(E\right)}{\rho \nu_{s}^{2}\hbar }\left\{{\left[1-\frac{\alpha E}{1+2\alpha E}\left(1-\frac{D_{v}}{D_{A}}\right)\right]}^{2}\right.\left.-\frac{8}{3}\frac{\alpha E(1+\alpha E)}{{\left(1+2\alpha E\right)}^{2}}\frac{D_{v}}{D_{A}}\right\},$$

where 
α
 describes the conduction band shape as in Equation (Equation 5), and 
ρ
 and 
$\nu_{s}$
 are the crystal’s density and speed of sound. In Si_x_Ge_1-x_, these have values of 
$\rho =2329+3493x-499x^{2}$
 kg/m^3^ and 
$\nu_{s}=\sqrt{(}B/\rho )$
, where *B* is bulk module which is given by 
B=98−23x
 GPA, with x is the atomic fraction of Ge [48]. The terms 
$D_{v}$
 and 
$D_{A}$
 are the electron and hole deformation potentials and are equal to 2.94 eV and 9.5 eV, respectively [49].

For strongly-screened Coulomb scattering that occurs when the carrier concentration is high, the electron scattering due to ionized impurities is given by [59]

(8)
$$\tau_{i}{\left(E\right)}^{-1}=\frac{\hbar }{\pi N_{i}{\left(\frac{e^{2}L_{D}^{2}}{4\pi \epsilon \epsilon_{o}}\right)}^{2}D\left(E\right)},$$

with 
$N_{i}$
 being the concentration of ionized impurities which we assume to be equal to the carrier concentration, 
$N_{i}=n_{c}$
. The terms 
ϵ
 and 
$\epsilon_{o}$
 are the relative and vacuum permittivity, with the former represented well with 
ϵ=11.7+4.5x
 in Si_x_Ge_1-x_ alloys [48]. The term 
$L_{D}$
 in Equation (Equation 8) is the Debye length, which in doped semiconductors has the generalized form of [60]

(9)
$$L_{D}=\frac{e^{2}N_{c}}{4\pi \epsilon \epsilon_{o}k_{B}T}\left[F_{-\frac{1}{2}}\left(\eta \right)+\frac{15\alpha k_{B}T}{4}F_{\frac{1}{2}}\left(\eta \right)\right],$$

where 
$N_{c}=2{\left(\frac{m^{*}k_{B}T}{2\pi \hbar^{2}}\right)}^{\frac{3}{2}}$
. We modeled the temperature dependence of the conduction band effective mass, 
$m^{*}$
, as 
$m^{*}\left(T\right)=m_{o}^{*}\left(1+5\alpha k_{B}T\right)$
[61]. The term 
$m_{o}^{*}$
 is equal to 
$0.28m_{o}$
, where 
$m_{o}$
 is the free electron rest mass as in Equation (Equation 5). The effective mass is temperature-dependent because the different sampling of the conduction band curvature as the Fermi window increases with temperature.

The rate of electron scattering due to the disordered arrangement of Ge atoms on the Si lattice is modeled as [62]

(10)
$$\tau_{a}{\left(E\right)}^{-1}=0.75\frac{x\left(1-x\right)3a^{3}\pi^{3}U_{A}^{2}m^{{*}^{\frac{3}{2}}}\sqrt{E}}{8\sqrt{2}\pi^{2}\hbar^{4}},$$

where x is the atomic fraction of Ge, *a* is the lattice parameter given as 
$a=5.431+0.2x+0.027x^{2}$
 [48] (5.47 Å for Si_0.8_Ge_0.2_). The term 
$U_{A}$
 is the alloy scattering potential and is equal to 0.7 eV for Si_0.8_Ge_0.2_ [49].

The three electron scattering terms above are sufficient to model single crystal Si_x_Ge_1-x_ with no porosity. We have validated this model against a set of phosphorous-doped Si_0.7_Ge_0.3_ experiments reported by Vining [63]. Figure 5 shows the comparison of the model prediction of electrical conductivity and Seebeck coefficient with the experimental data for Si_0.7_Ge_0.3_ with three different doping concentrations. We compared the results up to 1000 K. The model is in good agreement with experimental data in the whole range of temperature for the electrical conductivity. The prediction for *S* is less accurate, showing a small systematic underestimate of the Seebeck coefficient due to its sensitivity to the band shape and doping concentration. We remark that Vining only reported a single carrier concentration for each sample, so we had to assume that carrier concentration is constant across the span of temperature; however, it is likely that carrier concentration changes with temperature. The transport model was implemented as part of a python package called Thermoelectric.py that has been made available for public use at the GitHub repository in reference [64]. The Thermoelectric.py has been validated against experimental measurements in nanostructured Si containing a dispersion of SiC inclusions for which we have accurate and temperature resolved carrier concentration data, and these results will be in a forthcoming work.

In the nanostructured Si_0.8_Ge_0.2_ of interest in this study, there are two additional electron scattering processes that arise as a result of the morphology: electron scattering at grain boundaries, and scattering from pores. The rate of electron momentum relaxation due to elastic scattering from a uniform dispersion of pores can be modeled as [19]

(11)
$$\tau_{np}{\left(s\right)}^{-1}=\frac{N}{8\pi^{3}}\int SR_{kk^{\prime }}\left(1-\cos\left(\theta_{kk^{\prime }}\right)\right)dk^{\prime }.$$


Here, *N* is the number density of pores, and the term 
$SR_{kk^{\prime }}$
 is the rate of transition of an electron from an initial state with wave vector *k* and energy *E* to a state 
$k^{\prime }$
 with energy 
$E^{\prime }$
 due to a single pore. The 
$1-\cos(\theta_{kk^{\prime }})$
 term accounts for the change in momentum that accompanies this transition, with 
$\theta_{kk^{\prime }}$
 being the angle between initial and scattered wavevectors. For a time-invariant potential, the transition rate 
$SR_{kk^{\prime }}$
 is given by Fermi’s golden rule, 
$SR_{kk^{\prime }}=\frac{2\pi }{\hbar }{\left|M_{kk^{\prime }}\right|}^{2}\delta \left(E-E^{\prime }\right)$
, where the matrix element operator 
$M_{kk^{\prime }}$
 describes the strength which the pore couples the initial and final states and the number of ways the transition can occur, and 
δ
 is the Dirac delta function. For Bloch waves, 
$M_{kk^{\prime }}$
 is given by the integral of the overlap of the initial and final state with the pore potential 
U(r)
 so that [65]

(12)
$$M_{kk^{\prime }}=\int e^{i(k^{\prime }-k).r}U\left(r\right)dr.$$


For energy-conservative (elastic) scattering between eigenstates with the same energy Equation (Equation 11) can be recast as a surface integral over the isoenergetic k-space contour 
Γ
 that satisfies 
$E\left(k^{\prime }\right)=E\left(k\right)$


(13)
$$\tau_{np}^{-1}\left(s\right)=\frac{N}{{\left(2\pi \right)}^{2}\hbar }\oint_{\Gamma }\frac{{\left|M_{kk^{\prime }}\right|}^{2}}{\nabla E(k^{\prime })}\left(1-\cos\theta \right)dS\left(k^{\prime }\right),$$

where 
dS
 is the incremental area of the isoenergetic k-space surface. In most indirect bandgap semiconductors, such as Si_0.8_Ge_0.2_, the contours of isoenergy states near to conduction band valley have an ellipsoidal shape in momentum space that can be approximated as 
$E\left(k\right)=\hbar^{2}[\left(\frac{{\left(k_{l}-k_{ol}\right)}^{2}}{2m_{l}^{*}}+\frac{{\left(k_{t}-k_{ot}\right)}^{2}}{m_{t}^{*}}\right]$
, where 
E(k)
, 
$k_{o}=\left(k_{ol},k_{ot},k_{ot}\right)$
, 
$m_{l}^{*}$
, 
$m_{t}^{*}$
 are energy level from conduction band edge, the location of the conduction band minimum, longitudinal and transverse effective masses, respectively. We used 
$m_{l}^{*}=0.98m_{o}$
, 
$m_{t}^{*}=0.19m_{o}$
 where 
$m_{o}$
 is free electron rest mass, and 
$k_{o}=2\pi /a\left(0,0,0.85\right)$
, where *a* is the lattice parameter. The pore potential, 
U(r)
, in Equation (Equation 12) is assumed to be

(14)
$$U\left(r\right)=\left\{\begin{array}{cc} U_{o} & \mathrm{for}\;r\;\mathrm{inside}\;\mathrm{the}\;\mathrm{pore}\\ 0 & \mathrm{otherwise} \end{array}\right.,$$

where U_o_ = 4.05 eV is the electron affinity of bulk Si_0.8_Ge_0.2_. For infinitely-long cylindrical pores with radius 
$r_{o}$
, and aligned along the axis parallel to *z*, this gives the scattering matrix element operator

(15)
$$M_{kk^{\prime }}^{cylinder}=2\pi r_{o}U_{o}l_{z}\left(\frac{J_{1}\left(r_{o}q_{r}\right)}{q_{r}}\right)\delta_{k}\left(q_{z}\right).$$


In this equation, 
$q=k-k^{\prime }$
 is the scattering vector, and 
$q_{z}$
 and 
$q_{r}$
 are the components of *q* parallel and perpendicular to the cylinder axis. The term 
$\delta_{k}$
 is the Kronecker delta function, and 
$J_{1}$
 is the first-order Bessel function of the first kind, and 
$l_{z}$
 is the pore’s length perpendicular to the transport direction. We have previously computed the scattering matrix operators for pores with rectangular and triangular cross-sections and these can be found in reference [66]. The number density of pores is related to porosity, 
ϕ
, and the pore size through the relationship 
$N=\phi /V_{p}$
, where 
$V_{p}$
 is the volume of the pores.

A similar use of Fermi’s Golden rule can be used to model the rate of electron scattering by grain boundaries. Minnich et al. have suggested that grain boundaries provide a scattering potential of magnitude 
$U_{GB}$
 that decays away from the grain boundary over distance 
$z_{o}$
 [47]. From this, they derived the scattering operator matrix element for a small disc of grain boundary with radius 
$r_{o}$
 as

(16)
$$M_{kk^{\prime }}=4\pi U_{g}\left[\frac{z_{o}}{1+{\left(q_{z}z_{o}\right)}^{2}}\right]r_{o}^{2}\left[\frac{J_{1}\left(q_{r}r_{o}\right)}{q_{r}r_{o}}\right],$$

where 
$q_{r}$
 and 
$q_{z}$
 are the components of the scattering vector *q* that are parallel and perpendicular to the boundary, respectively. The scattering potential, 
$U_{GB}$
, is defined as

(17)
$$U_{GB}\left(r\right)=\left\{\begin{array}{cc} U_{g}e^{\frac{-|z|}{z_{o}}} & r<r_{GB}\\ 0 & \mathrm{otherwise} \end{array}\right..$$


In this equation, 
$z_{o}$
 is a constant related to the thickness of the depletion region at the grain boundary, and 
$U_{g}$
 was proposed to be 
$U_{g}=\frac{e^{2}N_{t}^{2}}{8\epsilon \epsilon_{o}N_{i}}$
. Here, 
ϵ
 is the permittivity, and 
$N_{t}$
 is the number density per area of electron traps in the depletion region. To compute the total scattering rate from all boundaries, the number density of grain boundary scattering centers is defined as 
$N=4f/(l_{g}r_{0}^{2})$
, where 
0<f<1
. Unfortunately, exact values of 
$r_{o}$
, 
$z_{o}$
, *f*, 
$N_{t}$
 are unknown. In this manuscript, we use the values proposed by Minnich et al. in their original paper on Si_0.8_Ge_0.2_ (
$r_{o}=1$
 nm, 
$z_{o}=2$
 nm, 
f=0.7
, 
$N_{t}=10^{13}$
 1/cm^2^, and we refer the reader to their work for the full details of the approach [47].

Figure 6 shows the variation in the electron lifetimes versus energy for the different scattering processes described above in Si_0.8_Ge_0.2_ doped to a carrier concentration of 10^20^ 1/cm^3^ at 500 K. At this doping level and temperature, impurity scattering is the strongest scattering process for low energy electrons, while high energy electrons are predominantly scattered by phonons; however, alloy, impurity, and phonon scattering all make a non-negligible contribution to the total rate of scattering. The scattering time due to grain boundaries in polycrystalline material with effective grain size 50 nm, and the scattering time due to 20 nm spaced cylindrical pores (for 
ϕ=25%
) are also plotted in Figure 6. These two scattering terms have a next to zero effect on total lifetime. This is an insightful result, showing the systematic difference between discrete and extended pores on electron scattering processes. We have studied the effect of discreet (e.g., spherical) pores, on electronic coefficients in a recent paper [66]. There, we have shown that the low-energy scattering by pores induced a strong filtering effect that considerably enhances the Seebeck coefficient and thus mitigates the effect of nanoscale porosity on the thermoelectric power factor of dielectrics. This phenomenon is also observed in other works [67]. However, for the extended pores considered here, scattering is only possible into states with the same component of wavevector along the pore axis, i.e., 
$q_{z}=0$
. This condition, combined with the isoenergetic constraint, reduces the scattering integral to an elliptical line, drastically reducing the number of states that can participate in scattering, and means that the extended pores cause next to no change in the electron momentum along the axis of the pores.

There is one final adjustment that must be made to the electrical transport model for the case of porous Si_x_Ge_1-x_. Although pores do not change the local material properties such as carrier concentration, the density of states, or Fermi energy, away from the pores, they do change the volume-averaged carrier concentration due to the reduction in the volume-averaged density of states. This will impact the conductivity, and thus the effective electrical conductivity of porous materials is modeled as 
$\sigma =\left(1-\phi \right)\sigma_{\mathrm{np}}$
. This change does not affect the Seebeck coefficient since it describes the relationship between two intensive quantities and so the changes in the density of state cancel out for the denominator and numerator in Equation (Equation 3). This means that the extended pores lower the power factor as 
$PF=\left(1-\phi \right)PF_{\mathrm{np}}$
.

## 4. Thermoelectric 
ZT
 of Nanoporous Polycrystalline Si_0.8_Ge_0.2_

To compute the total 
ZT
 of Si_x_Ge_1-x_ we combine the computed electrical and phonon transport properties described in the sections above with the electronic contribution to thermal conductivity computed using Equation (Equation 4). While phonons are the main contributors to thermal conductivity in crystalline dielectrics, in nanoengineered semiconductors where fine-grain boundaries significantly suppressed lattice thermal conductance, the electron contribution to heat conduction is considerable—this is of especial importance for designing TEs for high temperature working conditions, where maximum PF takes place at higher carrier concentrations, as can be seen in Figure A2 in Appendix A.

Figure 7 shows the best 
ZT
 performance that could be obtained by tuning the carrier concentration at each temperature, along with the corresponding optimal carrier concentration. It can be seen that both the addition of grain boundaries and nanopores produce a significant improvement in 
ZT
 with the grain boundaries having the stronger effect. The enhancements are not additive, so that there is little additional benefit to adding 5% porosity to polycrystalline Si_0.8_Ge_0.2_, but there is a significant gain to be had by adding 25% porosity. Most importantly, with the addition of nanostructure, the carrier concentration at which peak 
ZT
 occurs is reduced. Pores and grain boundaries have a negligible effect on electron scattering, but pores reduce the overall density of carriers reducing electrical conductivity. With the combination of pores and grain boundaries, phonon conductivity can be sufficiently suppressed so that the electric heat transport becomes significant. In this regime, 
ZT
 can be further enhanced by reducing the carrier concentration to reduce the electrical conductivity and electronic thermal conductivity while increasing the thermopower.

The solubility of P in Si at 1000 K is 10^21^ Atoms/cm^3^ [68], and the electrically active fraction of that is significantly lower. The solubility of P in Si_0.8_Ge_0.2_ is around half that of P in Si [69]. The carrier concentrations required to obtain peak 
ZT
 in single-crystal Si_0.8_Ge_0.2_ are likely to require a supersaturated concentration of P. This not only requires heating additional processing steps to achieve, but it is also easily destroyed during service if the material is inadvertently heated to a point when the dopant becomes mobile. With the addition of nanostructuring, Si_0.8_Ge_0.2_ requires only half the carrier concentration to obtain peak 
ZT
, making the material easier to process and more thermally robust.

## 5. Conclusions

To summarize, we have used a quasi-ballistic semiclassical Boltzmann transport model to elucidate the effect of extended nanopores with different shapes on the thermoelectric performance of Si_0.8_Ge_0.2_-based TE materials. We have shown that, while the pristine Si_0.8_Ge_0.2_ alloys’ thermal conductivity varies from about 12.5 W/mK at 200 K down to 5.4 W/mK at 1300 K, only 5% porosity of extended pore (100 nm spacing) can lower the conductivity to around 3 W/mK. Further increasing porosity to 25% lowers thermal conductivity to less than 1.7 W/mK. The porous alloys show very weak dependency on temperature as the rate of scattering of heat-carrying phonons by the phonon bath is superseded by scattering at interfaces. We further evaluated the effect of porosity on polycrystalline Si_0.8_Ge_0.2_ with effective grain sizes from 10 nm up to 200 nm. Cylindrical pores with 100 nm spacing reduced the thermal conductivity by more than 40% compared to the polycrystalline material with 50 nm grains but no pores. The importance of pores’ shape on thermal conductivity for three different geometries (cylindrical, cubic, and triangular prism) is studied through BTE simulations and phonons’ line-of-sight. Both methods accentuate that dielectrics containing triangular prism pores show the lowest thermal conductivity among the studied shapes. We further compared mode-resolved thermal conductivity across the frequency spectrum for cylindrical and triangular prism pores. We have also modeled the electron–pore scattering rate. The model demonstrated that electron has very weak coupling with extended pores with infinite length perpendicular to transport direction and therefore the changes in the power factor are only due to changes in the volume-averaged density of state and thus independent of pores’ shape. These predictions (both thermal and electrical) are in agreement with experimental measurements [23]. The model shows that introducing 20 nm spacing cylindrical nanopores in polycrystalline Si_0.8_Ge_0.2_ with 20 nm nanograins thermoelectrics can further improve the 
ZT
 up to 20%.

## Figures and Tables

**Figure 1 nanomaterials-11-02591-f001:**
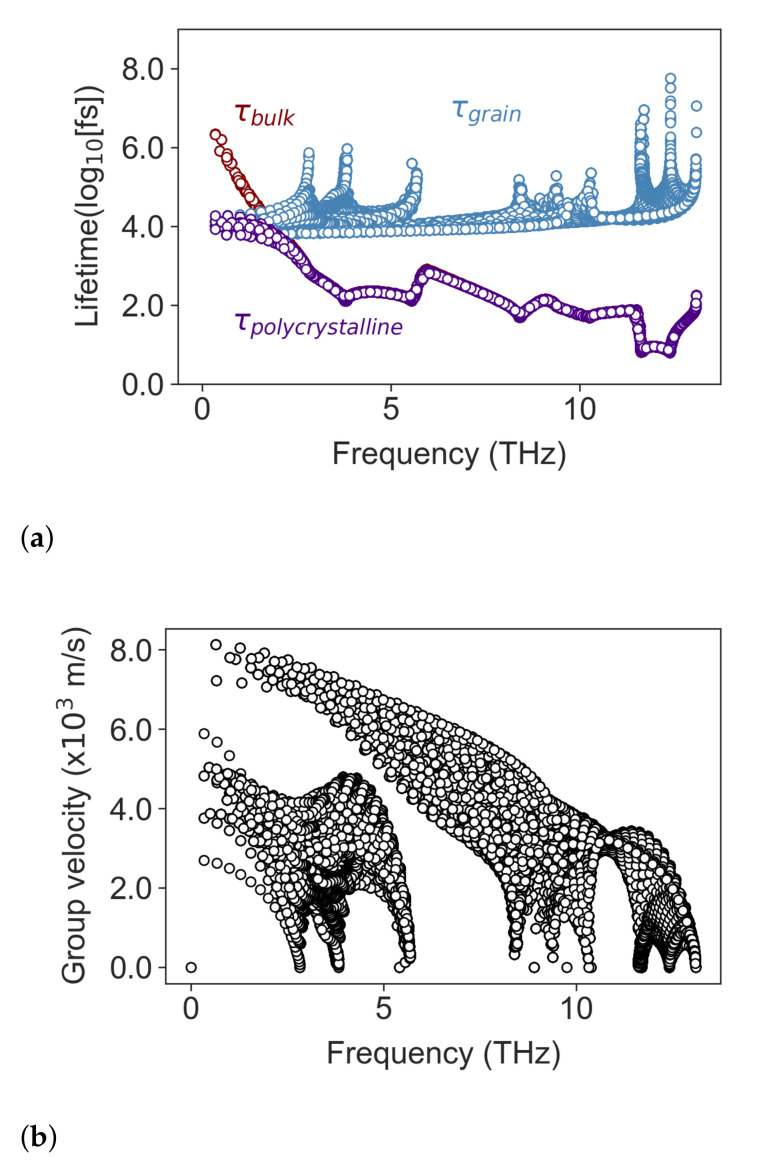
(**a**) Phonon lifetimes vs. frequency. The lifetime 
$\tau_{bulk}$
 due to phonon–phonon and phonon–alloy scattering processes in single crystal Si_0.8_Ge_0.2_ at 500 K is plotted in red. The lifetime 
$\tau_{grain}$
, due to grain boundary scattering in a microstructure with 
$l_{g}=50$
 nm is plotted in blue, and the total lifetime in from the combination of 
$\tau_{bulk}$
 and 
$\tau_{grain}$
 is plotted in purple. (**b**) The phonon group velocity used in the calculation of 
$\tau_{grain}$
.

**Figure 2 nanomaterials-11-02591-f002:**
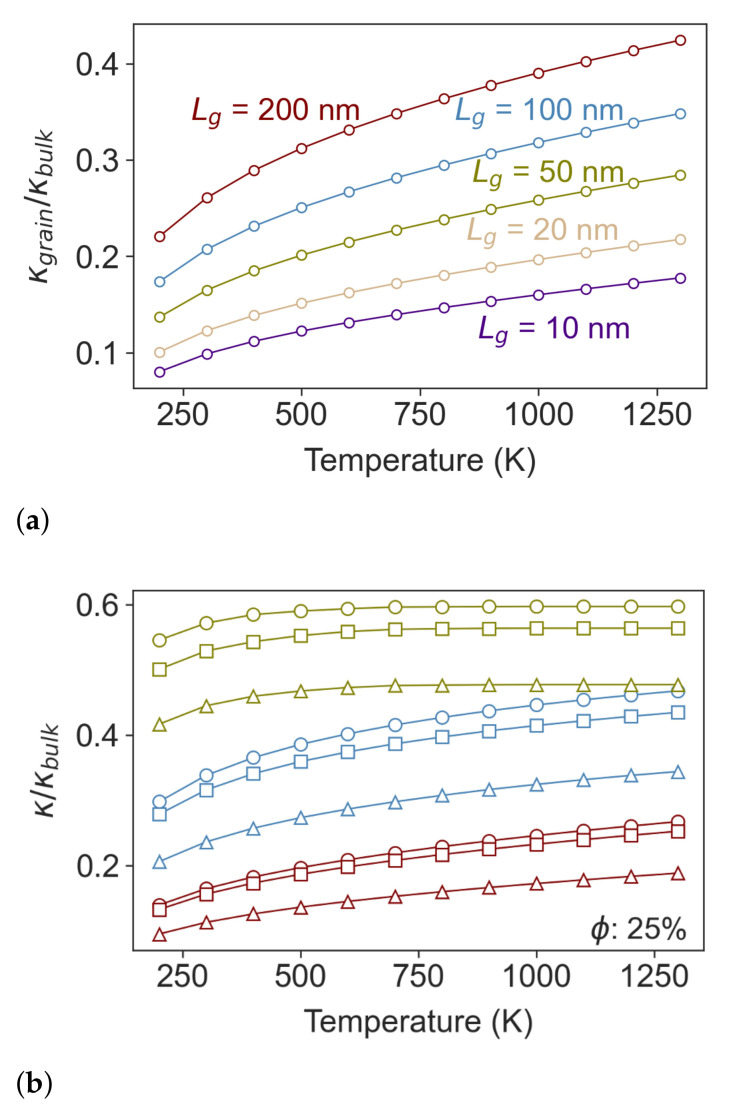
(**a**) Thermal conductivity of polycrystalline Si_0.8_Ge_0.2_ relative to the conductivity of single-crystal material at the same temperature. The red, blue, green, gold and purple plots are for effective grains sizes of 200 nm, 100 nm, 50 nm, 20 nm and 10 nm respectively. (**b**) Reduction in thermal conductivity of single-crystal Si_0.8_Ge_0.2_ alloy due to the addition of nanoscale porosity. The effects from pores with circular, square and triangular cross-section are plotted using markers of the same shape. The porosity is 
ϕ=0.25
 and the red, blue and green lines are for pore–pore distances of 0.1 
μ
m, 1 
μ
m and 20 
μ
m, respectively.

**Figure 3 nanomaterials-11-02591-f003:**
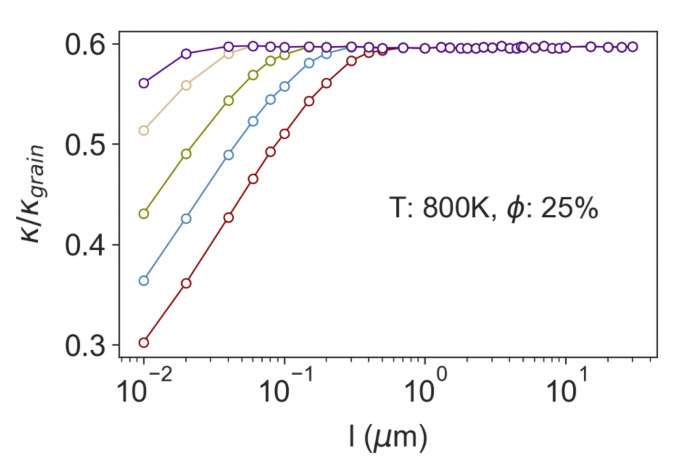
Thermal conductivity of polycrystalline Si_0.8_Ge_0.2_ containing cylindrical nanopores compared to the same polycrystalline material without pores. The thermal conductivity reduction is plotted vs. pore spacing, *l*, for material with effective grain sizes of 200 nm (red), 100 nm (blue), 50 nm (green), 20 nm (brown) and 10 nm (purple). For all cases, the pore fraction is 
ϕ=25%
.

**Figure 4 nanomaterials-11-02591-f004:**
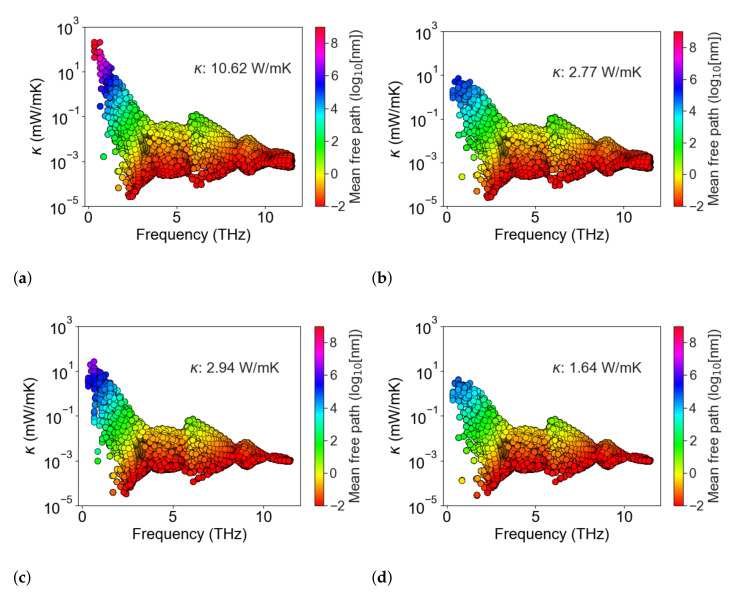
The per-mode thermal conductivity plotted vs. mode frequency in Si_0.8_Ge_0.2_ at 300 K. Plot (**a**) shows a monolithic single crystal, (**b**) shows a bulk polycrystalline material with effective grain size of 200 nm, (**c**) shows a single crystal containing cylindrical pores with 500 nm pore spacing, and (**d**) shows the same polycrystalline materials as in (**b**) with the addition of the cylindrical pores of (**c**). For each point, the mode’s mean-free path is indicated by the marker color using a log color scale.

**Figure 5 nanomaterials-11-02591-f005:**
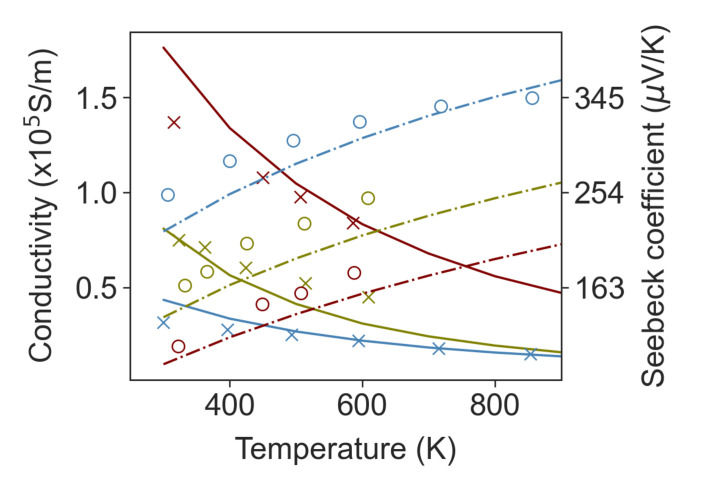
Comparison of the model prediction (lines) of electrical conductivity and thermopower in Si_0.7_Ge_0.3_ with experimentally reported results (markers). The measured and predicted electrical conductivity are shown with crosses and solid lines, respectively, and the measured and predicted Seebeck coefficients with circles and dashed lines. The data is for three different doping levels that have carrier concentrations of 
$1.45\times 10^{20}$
 1/cm^3^ (red), 
$6.75\times 10^{19}$
 1/cm^3^ (blue) and 
$2.2\times 10^{19}$
 1/cm^3^ (green). The experimental data are taken from reference [63]. The overall agreement is good, although the model gives a small systematic underestimate of the Seebeck coefficient.

**Figure 6 nanomaterials-11-02591-f006:**
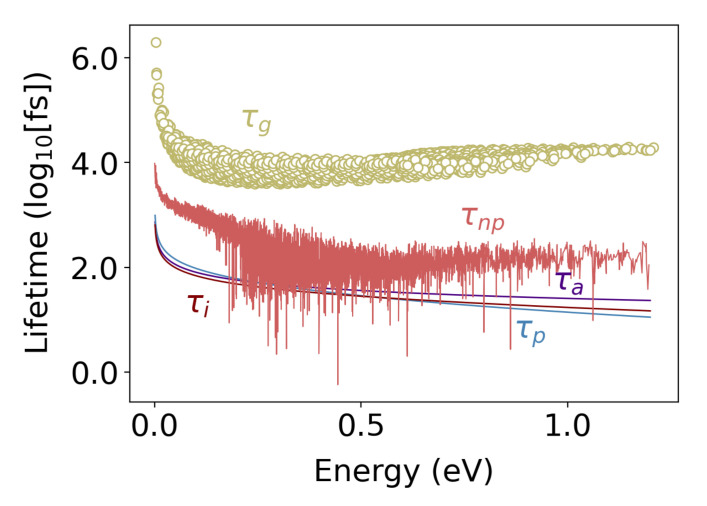
Electron lifetime for the different scattering mechanisms in Si_0.8_Ge_0.2_ at 500 K with a carrier population of 10^20^ 1/cm^3^. In low energy states, electron impurity is the strongest scattering term. For higher energy levels, electron–phonon scattering is the main source of scattering. The electron–grain boundary (
$l_{g}=50$
 nm) and electron–pore (pore–pore spacing of 20 nm) for 25% porosity are two additional scattering terms in polycrystalline porous Si_0.8_Ge_0.2_ that are shown in green and light red, respectively.

**Figure 7 nanomaterials-11-02591-f007:**
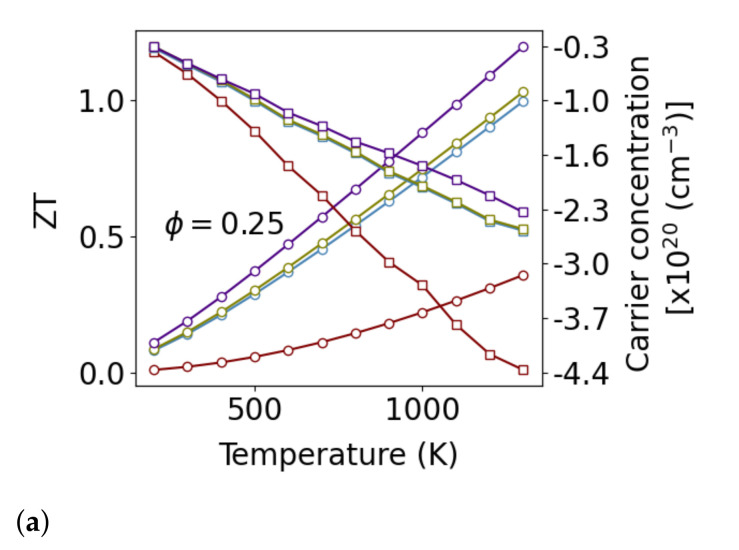
Plots of the maximum 
ZT
 that can be obtained by tuning the carrier concentration at each temperature. The data plotted with circles is the 
ZT
 and corresponds to the left hand axis, while data plotted with squares is the carrier concentration that produces the best 
ZT
. The red line is for monolithic single crystal Si_0.8_Ge_0.2_, the same material containing porosity 
ϕ
 in the form of cylindrical pores with a 20 nm spacing in blue, and polycrystalline Si_0.8_Ge_0.2_, with 20 nm grain size in green, and polycrystalline material with the 20 nm grain size and pores with a 20 nm spacing is plotted in purple. The top plot (**a**) shows 25% porosity, while plot (**b**) shows 5% porosity.

## Data Availability

The data that support the findings of this study are available from the corresponding author upon reasonable request.

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
