# Peer review of "Enhanced Thermoelectric Performance of Polycrystalline Si0.8Ge0.2 Alloys through the Addition of Nanoscale Porosity"

_nanomaterials, 2021, doi:10.3390/nano11102591_

Round 1

Reviewer 1 Report

In this article,authors used quasiballistic semiclassical Boltzmann transport model to elucidate the effect of extended nanopores with different shapes on thermoelectric performance of Si0.8Ge0.2 based TE materials. The influence of the pore shape, the pore-pore distances and other factors on the thermal and electrical properties has been studied. In short, this is a relatively systematic article. The comments are:

(1) Is phonon-phonon scattering only the Umklapp scattering? Does the Normal scattering need to be considered?

(2) In this model, how is the effect of pore shape on the thermal and electrical properties reflected?

(3) Can the correctness of the model be verified through experiments?

Reviewer 2 Report

 I would like to ask the following:

  1. Cited here the paper “Effects of nanoscale porosity on thermoelectric properties of SiGe” published by Ref 14 is very important for the current topic. Moreover, it was written and shown by Ref 14 “Despite a thermal conductivity reduction, it has been experimentally observed that the porous nanograined materials have lower thermoelectric figure of merit than their bulk counterpart due to significant reduction in the electrical conductivity”.
    Moreover, similar conclusion “The presence of hollow pores of different size scales randomly distributed throughout the matrix thermoelectric material leads to the enhancement of the thermopower and the reduction of the electrical conductivity relative to their values in the bulk material with zero porosity” was done in the paper of Tarkhanyan  “Thermoelectric power factor in nano- to microscale porous composites”.
    However, the authors of the current manuscript have shown the opposite results between single crystal or bulk SiGe and polycrystalline materials without explanation (see Fig. 7 for monolithic single crystal or 10b for bulk). Discussion/comparison must be present.

2. There is missing the comparison of obtained theoretical results with experimental papers that described influence of nanopores, defects, inclusion (such as graphene, i.e. https://doi.org/10.3390/c7020037), etc.

3. Why 500 K was selected for Figure 6: the variation in the electron lifetimes versus energy for the different scattering processes?

Round 2

Reviewer 2 Report

It can be published